# Bridging the Modality Gap: Dimension Information Alignment and Sparse Spatial Constraint for Image-Text Matching

Submission Id: 3911

## ABSTRACT

Many contrastive learning based models have achieved advanced performance in image-text matching tasks. The key of these models lies in analyzing the correlation between image-text pairs, which involves cross-modal interaction of embeddings in corresponding dimensions. However, the embeddings of different modalities are from different models or modules, and there is a significant modality gap. Directly interacting such embeddings lacks rationality and may capture inaccurate correlation. Therefore, we propose a novel method called DIAS to bridge the modality gap from two aspects: (1) We align the information representation of embeddings from different modalities in corresponding dimension to ensure the correlation calculation is based on interactions of similar information. (2) The spatial constraints of inter- and intra-modalities unmatched pairs are introduced to ensure the effectiveness of semantic alignment of the model. Besides, a sparse correlation algorithm is proposed to select strong correlated spatial relationships, enabling the model to learn more significant features and avoid being misled by weak correlation. Extensive experiments demonstrate the superiority of DIAS, achieving 4.3%-10.2% rSum improvements on Flickr30k and MSCOCO benchmarks.

## CCS CONCEPTS

• **Information systems → Information retrieval**.

## KEYWORDS

Image-text Matching, Information Aligning, Spatial Constraint, Sparse Algorithm

## 1 INTRODUCTION

Image-text matching is a fundamental task in computer vision (CV) and natural language processing (NLP), providing support for applications such as image captioning [12, 38], text retrieval [15], and text-to-image generation [8, 18]. This task aims to discover semantic correlations between images and text, and bridge the semantic gap between these two heterogeneous modalities. The key challenge lies in adjusting embeddings by utilizing matched and unmatched relationships between images and texts to achieve high-quality semantic alignment.

The matching process typically requires matching with embeddings constructed from images and texts. The existing methods can be roughly divided into two categories: global and local [5, 20]. Global-based matching extracts and interacts with global embeddings from the whole images and texts to calculate correlations [2, 24]. Local-based matching adopts a fine-grained approach, which extracts local embeddings from image regions and text words usually obtains better performance [1, 23, 41]. They all aim at aligning semantics by computing and adjusting the correlation between embeddings of different modality, which involves interaction of corresponding dimensions. For example, cosine similarity [28] calculates the correlation between two embeddings in each dimension. However, the embeddings generally come from different models or modules, resulting in significant differences in information representation of each dimension. For instance, the image embeddings represent color information in a certain dimension, while the text embeddings may represent the information of a word in the corresponding dimension. Note that the corresponding dimension may not necessarily be in the same column of embeddings. This is known as the modality gap problem. The cross-modal interaction of such embeddings lacks rationality and potentially lead to inaccurate correlation calculation.

To enhance the rationality and effectiveness of cross-modal interaction, we propose a novel image-text matching method based on **D**imensional **I**nformation **A**lignment and **S**parse Spatial Constraint (DIAS), aiming to bridge the gap between image and text modalities from two perspectives:

(1) To ensure the rationality of correlation calculation, we enhance the correlation of the embeddings from different modalities in corresponding dimension. In subsequent processes, the interaction involves the relevant information of embeddings in their corresponding dimensions. Emphasizing only the correlation of dimensions may lead to feature redundancy, where each dimension provides similar information and lacks discriminative features. Feature redundancy can cause overfitting, reducing the generalization ability of models. Therefore, we enhance the independence of non-corresponding dimensions by reducing the correlation of them, to ensure the amount of information contained in embeddings.

(2) Most existing methods primarily focus on constraining the relationships between matched image-text pairs, with weaker emphasis on unmatched pairs. This can lead to suboptimal performance in semantic alignment. More importantly, the relationship of matched pairs is cross modal constraints, and their effectiveness is significantly affected by the modality gap. We augment existing constraints by introducing spatial inter- and intra-modalities constraints for unmatched pairs. The inter-modality constraint refers to promoting semantic consistency by requiring distance consistency between inter-modality unmatched pairs. As shown in Fig. 1(a), the distance between image $i$ and text $j$ is constrained to be consistent

with the distance between image $j$ and text $i$. The intra-modality constraint refers to emphasizing spatial structure consistency by requiring distance consistency between unmatched pairs within each modality. As shown in Fig. 1(b), the distance between image $i$ and image $j$ is constrained to be consistent with the distance between text $i$ and text $j$. However, these two types of constraints assume the spatial relationships between images and texts exhibit symmetry, which is not always valid. Strictly following these constraints may lead to the model learning inaccurate features. Therefore, we propose a sparse correlation algorithm to select strong correlation to sparsify spatial constraints, reducing the need for symmetry.

Specifically, DIAS first obtains local embeddings of image regions and text words, and calculates the correlations between them in all dimensions to construct the correlation matrix. Each value in the matrix means the correlation of the corresponding region (row) and word (column). To align the information of embeddings from different modalities, we propose a regularizer to increase the correlation values of corresponding dimensions. Meanwhile, the correlation values between non-corresponding dimensions are decreased to suppress feature redundancy. Then, DIAS aggregates and upgrates the local embeddings, and merges them into global embeddings by pooling. As correlations of local embeddings have been adjusted in the previous step, the construction of global embeddings becomes more reasonable. Subsequently, DIAS obtains the spatial distance between inter- and intra-modalities unmatched pairs, and further employs the proposed sparse correlation algorithm to select strong correlation from them. The proposed algorithm introduces conditional probabilities of instance correlation and adapts them into a sparse regularization term, enabling the model to automatically learn how to identify strong correlation for each instance. Finally, the selected spatial relationships are used as constraints, combined with the constraints between matched pairs to achieve semantic alignment.

Our contributions are summarized as follows:

(1) We propose a dimension information alignment method for embeddings of different modalities, aiming to enhance the rationality of cross-modal interaction and suppress feature redundancy.

(2) We introduce novel inter- and intra-modality constraints to ensure the effectiveness of semantic alignment.

(3) A sparse correlation algorithm is proposed to select strong correlated spatial relationships, reducing the need for symmetry of embeddings.

## 2 RELATED WORK

Based on the implementation of cross-modal interactions, the image-text matching methods can be broadly categorized into global-based matching and local-based matching method.

**Global-based matching.** The typical global methods involve obtaining global embeddings of images and texts, projecting them into a shared embedding space by two branches and aligning image-text semantic. A line of works focus on how to accurately describe correlations between global embeddings. Some studies [3, 6] focus on improving correlation algorithms. For example, Jiang [9] introduces the concept of geometric consistency to enhancing the constraint on image-text pairs. Additionally, some studies [10, 11, 13, 17, 32] propose complex models to construct more robust

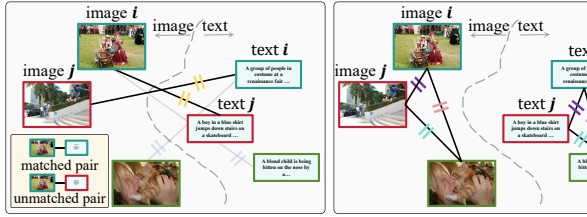

(a) Inter-modality distance consistency   (b) Intra-modality distance consistency

**Figure 1: Illustration of distance consistency.**

global embeddings. Especially in recent years, pre-trained networks [16, 25] with extensive resources enrich the information contained in global embeddings. However, these methods still follow the existing paradigm, assuming embeddings from different modalities interact with the same information during correlation computation. In contrast, we focus on aligning the information representation of embeddings to enhance the rationality of correlation computation.

**Local-based matching.** Learning semantic alignment between local embeddings from image regions and text words is popular and offers better interpretability compared to global methods. Karpathy [10] makes the first attempt to infer matching between regions and words by aggregating similarities across all regions and words to obtain the correlation between image and text. A line of works focuses on constructing thoughtful aggregation rules to find the important region-word pairs. Chen [1] proposes recurrent cross-attention to iteratively refine and elaborate shared semantics across different levels. Zhang [39] introduces negative-aware attention on unmatched pairs to enhance matching accuracy. Pan [22] considers that effective image-text semantic matching can be achieved solely by relying on the maximum region-word correlation and provides theoretical derivation. Another line of works focuses on exploiting more information. Wang [31] introduces scene graph during matching to enrich relationships between local embeddings. Additionally, the models combining consensus knowledge [30] and external pre-training knowledge [24, 33] have been employed to enhance the cross-modal alingment. However, they still rarely consider the differences of information representation in different dimensions caused by modality gap. As mentioned earlier, we bridge the modality gap by aligning information representation of embeddings.

## 3 METHODOLOGY

Considering effectiveness and interpretability, DIAS adopts the local-based matching method. In this section, we introduce the framework of local-based matching method (Sec. 3.1) and the details of DIAS. As shown in Fig.2, DIAS first perfroms dimension information alignment to adjust the information representation of the embeddings in different dimensions (Sec. 3.2). Then inter- and alities spatial constraints are introduced to suppress the influence of the modality gap (Sec. 3.3), and the sparse conrrelation algorithm is used to select the strong correlated spatial relationships (Sec. 3.4).

### 3.1 The Framework of Local-based Matching

Formally, given an image $\mathbf{V}$, we use Faster-RCNN [26] to extract the salient regions and obtain the local image embeddings $\mathbf{V} = \{\mathbf{v}_i | i \in [1, n_v], \mathbf{v}_i \in \mathbb{R}^d\}$ by the pre-trained ResNet-101 [7]. $\mathbf{v}_i$ is the

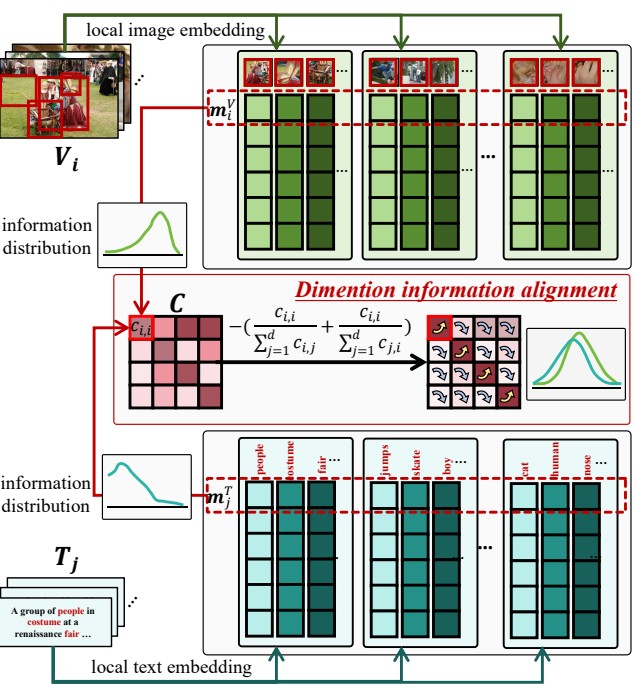

Figure 2: Overview of DIAS, which mainly contains two steps: local embedding interaction and global embedding interaction. Firstly, DIAS extracts features from image regions and text words to construct local embeddings, and perfroms dimension information alignment to adjust the information representation of the embeddings in different dimensions ($\mathcal{L}_{dim}$). Then, we aggregates local embeddings to construct global embeddings. Inter- and intra-modalities spatial constraints are obtained from distance relationship between global embeddings, to suppress the influence of the modality gap, and the sparse conrrelation algorithm is used to select the strong correlated spatial relationships ($\mathcal{L}_{inter}$ and $\mathcal{L}_{intra}$). Finally, the image-text relevance is inferred via a contrastive learning loss function ($\mathcal{L}_{loc}$).

local embeddings of $i$-th region. $n_v$ denotes the number of regions. Similarly, given text $\mathbf{T}$, we employ Bidirectional Gated Recurrent Units (BiGRU) [27] or BERT [4] to extract local text embeddings $\mathbf{T} = \{\mathbf{t}_j | j \in [1, n_t], \mathbf{t}_j \in \mathbb{R}^d\}$. $\mathbf{t}_j$ is the local embeddings of $j$-th words. $n_t$ denotes the number of words.

Local-based matching first conduct local embedding interaction to update local embeddings based on the correlation between regions and words. The updating of $\mathbf{v}_i$ can be described as follows:

$$\hat{\mathbf{v}}_i = \frac{\sum_{j=1}^{n_t} s_{i,j} \mathbf{t}_j}{\sum_{j=1}^{n_t} s_{i,j}}, \quad i \in [1, n_v] \tag{1}$$
$$s_{i,j} = \sigma_l(\mathbf{v}_i, \mathbf{t}_j)$$

Here $\hat{\mathbf{v}}_i$ represents the new local embedding. $\sigma_l(\cdot)$ is the correlation function for local embeddings. $s_{i,j}$ is the correlation value between $\mathbf{v}_i$ and $\mathbf{t}_j$. Then, local embeddings are transformed into global embeddings by pooling, formally as:

$$\hat{\mathbf{V}} = pool(\{\hat{\mathbf{v}}_i | i \in [1, n_v]\}) \tag{2}$$

Here $\hat{\mathbf{V}}$ is the global embedding of image $\mathbf{V}$. $pool(\cdot)$ means the pooling operation. Through the similar process, we can obtain the local embedding of word $\hat{\mathbf{t}}_j$ and global embedding of text $\hat{\mathbf{T}}$.

The correlation between image and text is obtained based on global embedding interaction. The triplet loss is the most commonly used method for achieving semantic alignment, and the objective

Figure 3: Illustration of dimension information alignment. We extract the dimension vector of each dimention, and construct the correlation matrix by calculating the correlation between dimension vectors from different modalities. The proposed regularizer is used on the correlation matrix to align information repersentaion of each dimension.

function can be expressed as:

$$\mathcal{L}_{loc} = [\alpha - \sigma_g(\hat{\mathbf{V}}, \hat{\mathbf{T}}) + \sigma_g(\hat{\mathbf{V}}, \hat{\mathbf{T}}^-)]_+ + [\alpha - \sigma_g(\hat{\mathbf{V}}, \hat{\mathbf{T}}) + \sigma_g(\hat{\mathbf{V}}^-, \hat{\mathbf{T}})]_+ \tag{3}$$

Here $\alpha$ means a margin parameter, $[\cdot]_+ = max(\cdot, 0)$. $\sigma_g$ is the correlation function for instances. $(\hat{\mathbf{V}}, \hat{\mathbf{T}})$ is a positive image-text pair, and $(\hat{\mathbf{V}}, \hat{\mathbf{T}}^-)$ and $(\hat{\mathbf{V}}^-, \hat{\mathbf{T}})$ are negative image-text pair in the batch. We use the distance-weighted sampling [35] for hard negative mining.

## 3.2 Dimension Information Alignment

The correlation calculation likes Eq.1 involves the cross-modal interaction in corresponding dimensions of embeddings. As mentioned earlier, due to the different sources, there are significantly differences in information representation of $\mathbf{v}_i$ and $\mathbf{t}_j$ in different dimensions. The interaction of them can result in calculation biases and lack of rationality. Thus, we propose a dimension information alignment method to align the information representation before the interaction by a regularizer. It can improves the correlation of $\mathbf{v}_i$ and $\mathbf{t}_j$ in corresponding dimensions. Meanwhile, to suppress feature redundancy that may occur during the alignment, the regularizer also reduces the correlation values between non-corresponding dimensions. Below is a detailed introduction to this process.

Assuming there are $N$ image-text pairs. As shown in Fig. 3, we first extract dimension vectors of all local embeddings, and integrate them into $\mathbf{m}^V = \{\mathbf{m}_i^V | i \in [1, d], \mathbf{m}_i^V \in \mathbb{R}^{N_V}\}$ and

$\mathbf{m}^T = \{\mathbf{m}_j^T | j \in [1, d], \mathbf{m}_i^T \in \mathbb{R}^{N_T}\}$, respectively. Here $\mathbf{m}_i^V$ contains the information distribution of all local image embeddings in $i$-th dimension, and $\mathbf{m}_j^T$ contains the information distribution of all local text embeddings in $j$-th dimension. The number of regions in different images and the number of words in different texts vary. So, we use $N_V$ and $N_T$ to represent the total number of regions and words, respectively. Then, we compute the correlation between $\mathbf{m}_i^V$ and $\mathbf{m}_j^T$, formally as:

$$c_{i,j} = \sigma_c(\mathbf{m}_i^V, \mathbf{m}_j^T), \quad i, j \in [1, d] \tag{4}$$

Here $c_{i,j}$ is the correlation value of $\mathbf{m}_i^V$ and $\mathbf{m}_j^T$. $\sigma_c$ denotes the correlation algorithm for dimension vectors. The correlation matrix $\mathbf{C} = \{c_{i,j} | i, j \in [1, d]\}$ can be obtained via Eq.4.

Then, we use a regularizer to improve the correlation of corresponding dimensions and reduce the correlation between non-corresponding dimensions. For ease of understanding, we assume the corresponding dimensions are at the same column of embeddings. It means the corresponding dimension of $\mathbf{m}_i^V$ is $\mathbf{m}_i^T$ and $c_{i,i}$ is the correlation value of them. The regularizer can be expressed as:

$$\mathcal{L}_{dim} = -\sum_{i=1}^{d} c_{i,i} + \sum_{i=1}^{d} \sum_{j=1, j \neq i}^{d} c_{i,j} \tag{5}$$

The first term of Eq.5 mainly aligns the corresponding dimension, and the second term misaligns the non-corresponding dimensions. The setting of this function is relatively intuitive, but it fails to account for the magnitude difference in rows or columns of $S$, potentially leading to computational bias. Therefore, we improve it to the following formula:

$$\mathcal{L}_{dim} = \sum_{i=1}^{d} -\left( \frac{c_{i,i}}{\sum_{j=1}^{d} c_{i,j}} + \frac{c_{i,i}}{\sum_{j=1}^{d} c_{j,i}} \right) \tag{6}$$

As shown in Eq.6, the regularizer increases the proportion of $c_{i,i}$ to corresponding rows and columns in $C$, avoiding the impact of inconsistent orders of magnitude.

After aligning the dimension information, the process of aggregating and upgrating the local embeddings in Eq.1 generates more reasonable correlations. Moreover, the information representation of $\hat{\mathbf{V}}$ and $\hat{\mathbf{T}}$ in the corresponding dimensions obtained by Eq.2 is also more similar.

## 3.3 Spatial Constraint

After obtaining the global embeddings $\hat{\mathbf{V}}$ and $\hat{\mathbf{T}}$, we calculate their correlation and use the loss function (Eq.3) to achieve semantic alignment. For each instance, the number of unmatched instances far exceeds the number of matched instances. Existing methods often impose stronger constraints on matched pairs and weaker constraints on unmatched pairs. For example, Eq.3 requires the correlation of matched pairs is greater than that of all unmatched pairs, while unmatched pairs only need to satisfy a threshold $\alpha$ smaller than that of matched pairs. To ensure the effectiveness of semantic alignment, we propose two spatial constraint regularizers to enhance the constraint on unmatched pairs, including inter- and intra-modalities constraints.

On the one hand, we aim to maintain semantic consistency by pursuing spatial distance consistency of inter-modality unmatched

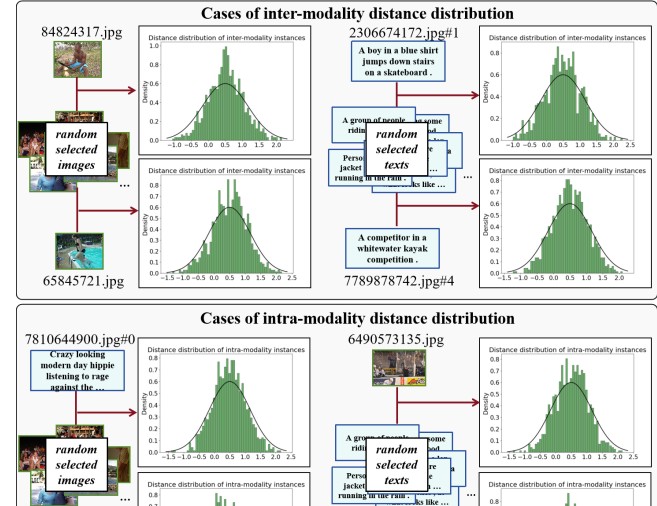

**Figure 4: The histogram statistics of spatial distance between instances within and across modalities. We randomly selected some images and texts to calculating their distance, and observe the distribution pattern. It can be observed that the inter- and intra-modalities distance distribution approaches a normal distribution. These embeddings used for computation are from the state-of-the-art method [37].**

pairs. Concretely, we compute the distance of all global embeddings between different modalities:

$$x_{i,j} = \sigma_x(\hat{\mathbf{V}}_i, \hat{\mathbf{T}}_j), \quad i, j \in [1, N] \tag{7}$$

Here $\hat{\mathbf{V}}_i$ is the global embedding of $i$-th image, and $\hat{\mathbf{T}}_j$ is the global embedding of $j$-th text. $N$ is the number of image-text pairs, and assuming the matched pair of $\hat{\mathbf{V}}_i$ is $\hat{\mathbf{T}}_i$. $\sigma_x$ is the distance function. $x_{i,j}$ is the spatial distance between $\hat{\mathbf{V}}_i$ and $\hat{\mathbf{T}}_j$. We combine $x_{i,j}$ to construct spatial matrix $\mathbf{X} = \{x_{i,j} | i, j \in [1, N]\}$. The regularizer for inter-modality unmatched pairs is as follwing:

$$\mathcal{L}_{inter} = ||\mathbf{L}_x||_2^2 = ||\mathbf{X} - \mathbf{X}^\top||_2^2 = \sum_{i=1}^{N} \sum_{j=1}^{N} (x_{i,j} - x_{j,i})^2$$

$$= \sum_{i=1}^{N} \sum_{j=1}^{N} (\sigma_x(\hat{\mathbf{V}}_i, \hat{\mathbf{T}}_j) - \sigma_x(\hat{\mathbf{V}}_j, \hat{\mathbf{T}}_i))^2 \tag{8}$$

Here $\mathbf{L}_x = |\mathbf{X} - \mathbf{X}^\top|$ is the inter-modality spatical matrix to be optimized. It can be observed that this regularizer imposes strong distance constraint only on unmatched pairs, which partially compensates for the shortcomings of Eq.3. The regularizer can effectively reduce the model's sensitivity and enhance its robustness and generalization when handling diverse modality data. But it still handles inter-modality embeddings, which are limited by modality gap.

So, on the other hand, we aim to maintain structure consistency of different modalities by pursuing spatial distance consistency of

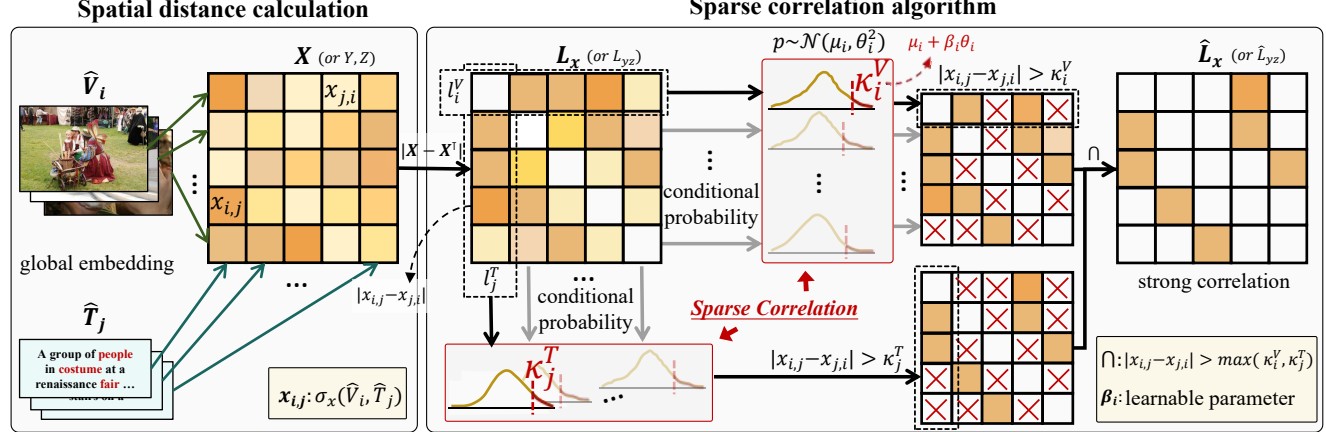

**Figure 5: Illustrasion for sparse correlation algorithm. We obtain the spatial matrix $\mathbf{L}_x$, and the model learns a soft-threshold based on the conditional probability to select strong correlation for each instance.**

intra-modality unmatched pairs. We compute the distance of all global embeddings in each modality:

$$y_{i,j} = \sigma_y(\hat{\mathbf{V}}_i, \hat{\mathbf{V}}_j), \quad i, j \in [1, N]$$
$$z_{i,j} = \sigma_z(\hat{\mathbf{T}}_i, \hat{\mathbf{T}}_j) \tag{9}$$

Here $y_{i,j}$ means the spatial distance between $\hat{\mathbf{V}}_i$ and $\hat{\mathbf{V}}_j$. $z_{i,j}$ means the spatial distance between $\hat{\mathbf{T}}_i$ and $\hat{\mathbf{T}}_j$. $\sigma_y$ and $\sigma_z$ are the distance functions of images and texts, respectively. We combine $y_{i,j}$ to construct $\mathbf{Y} = \{y_{i,j} | i, j \in [1, N]\}$ and combine $z_{i,j}$ to construct $\mathbf{Z} = \{z_{i,j} | i, j \in [1, N]\}$. The regularizer for intra-modality unmatched pairs is as follwing:

$$\mathcal{L}_{intra} = ||\mathbf{L}_{yz}||_2^2 = ||\mathbf{Y} - \mathbf{Z}||_2^2 = \sum_{i=1}^{N} \sum_{j=1}^{N} (y_{i,j} - z_{i,j})^2$$
$$= \sum_{i=1}^{N} \sum_{j=1}^{N} (\sigma_y(\hat{\mathbf{V}}_i, \hat{\mathbf{V}}_j) - \sigma_z(\hat{\mathbf{T}}_i, \hat{\mathbf{T}}_j))^2 \tag{10}$$

Here $\mathbf{L}_{yz} = |\mathbf{Y} - \mathbf{Z}^\top|$ is the inter-modality spatical matrix to be optimized. It can be observed that this regularizer constrains embeddings of different modalities to have the same spatial relationships, enhancing their consistency of spatial structure. Furthermore, it only processes embeddings within modalities and is not affected by modality gap. Even if the modality gap is not completely eliminated, this regularizer can still encourage our model to learn effective features.

## 3.4 Sparse Correlation Algorithm

The spatial constraints assume the spatial relationships between images and texts exhibit symmetry, but this assumption is not always valid. These inaccurate relationships can affect the performance of the model. Fig. 4 shows the distribution of spatial distance between instances within and across modalities. It can be observed that the relationships between instances are mostly weakly correlated, and these relationships have little effect on characterizing the spatial position of instances. Considering the effectiveness and efficiency, we propose a sparse correlation algorithm. This algorithm concentrates

spatial constraints on strong correlation relationships to capture more significant and important features. More importantly, this algorithm can reduce the need for embedding symmetry, making it more flexible.

The key issue is how to determine which instances exhibit strong correlations. The correlation distribution of different instances varies greatly, making it unsuitable to set a unified hard-threshold to distinguish strong and weak correlations. Therefore, we propose a sparse correlation algorithm to adaptively distinguish strong and weak correlations based on the situation of the instance itself. This algorithm builds conditional probabilities of correlation and uses them to obtain the soft-threshold, as shown in Fig. 5. Taking matrix $\mathbf{L}_x$ as example, we set its $i$-th row vector as $\mathbf{l}_i^V = \{l_{i,j}^V | j \in [1, N]\}$, and $l_{i,j}^V = |x_{i,j} - x_{j,i}|$. $l_i^V$ indicates the correlation between image $\mathbf{V}_i$ and all texts. Similar, we set the $j$-th column vecter of $\mathbf{L}_x$ as $\mathbf{l}_j^T = \{l_{j,i}^T | i \in [1, N]\}$, and $l_{j,i}^T = |x_{j,i} - x_{i,j}|$. To explicitly quantification, we represent the conditional probability of each image as:

$$p(l_i^V | l_j^T) = Sigmoid(-|x_{i,j} - x_{j,i}|), \quad j \in [1, N] \tag{11}$$

Here $p(l_i^V | l_j^T) \in [0, 1]$ represents the dependency degree of $\hat{\mathbf{V}}_i$ on $\hat{\mathbf{T}}_j$. A larger value of $p(l_i^V | l_j^T)$ indicates a stronger dependency. We expect the model to discover strong correlations for each image and text based on the latent semantics of $L_x$, to avoid interference from other weakly correlated instances and to be as concise as possible. Specifically, we observed that the histogram of the conditional probability $\{p(l_i^V | l_j^T)\}_{j=1}^d$ approximates a normal distribution, as shown in Fig. 4. Therefore, based on the statistical features of conditional probabilities, we can enable the model to learn a soft-threshold for distinguishing strong and weak correlations for each instance:

$$\kappa_i^V = \mu_i + \beta_i \cdot \theta_i \tag{12}$$

Here $\kappa_i^V$ is the soft-threshold of $\mathbf{l}_i^V$. $\mu_i$ and $\theta_i$ are the mean and standard deviation of the sampling probability values from $\{p(l_i^V | l_j^T)\}_{j=1}^d$, respectively. $\beta_i$ is a learnable parameter to adjust the sparse degree.

**Table 1: Comparisons with state-of-the-art methods on Flickr30k and MSCOCO 1K test-sets. *BUTD* represents using Faster-RCNN [2] to extract local image embeddings. *BiGRU* and *BERT* represent using BiGRU [27] or BERT [4] to extract local text embeddings. * denotes the ensemble results of two models. The bests are in bold.**

| Methods | Flickr30K | | | | | | | MSCOCO 1K | | | | | | |
| | IMG→TEXT | | | TEXT→IMG | | | rSum | IMG→TEXT | | | TEXT→IMG | | | rSum |
| | R@1 | R@5 | R@10 | R@1 | R@5 | R@10 | | R@1 | R@5 | R@10 | R@1 | R@5 | R@10 | |
| ***BUTD+BiGRU*** | | | | | | | | | | | | | | |
| GSMN*(2020)[20] | 76.4 | 94.3 | 97.3 | 57.4 | 82.3 | 89.0 | 496.8 | 78.4 | 96.4 | 98.6 | 63.3 | 90.1 | 95.7 | 522.5 |
| GPO(2021)[2] | 76.5 | 94.2 | 97.7 | 56.4 | 83.4 | 89.9 | 498.1 | 78.5 | 96.0 | 98.7 | 61.7 | 90.3 | 95.6 | 520.8 |
| MV(2022)[17] | 79.0 | 94.9 | 97.7 | 59.1 | 84.6 | 90.6 | 505.8 | 78.7 | 95.7 | 98.7 | 62.7 | 90.4 | 95.7 | 521.9 |
| NAAF*(2022)[39] | **81.9** | **96.1** | 98.3 | **61.0** | 85.3 | 90.6 | 513.2 | 80.5 | 96.5 | 98.8 | 64.1 | 90.7 | **96.5** | 527.2 |
| CHAN(2023)[22] | 79.7 | 94.5 | 97.3 | 60.2 | 85.3 | 90.7 | 507.8 | 79.7 | 96.7 | 98.7 | 63.8 | 90.4 | 95.8 | 525.0 |
| NUIF-d(2024)[37] | 81.8 | 94.7 | 97.6 | 59.4 | **85.6** | 91.1 | 509.3 | 80.6 | 96.3 | 98.8 | 64.7 | **91.4** | 96.2 | 528.0 |
| **DIAS(ours)** | 81.8 | **96.1** | **98.6** | 60.7 | 84.9 | **91.3** | **513.4** | 81.3 | 96.8 | 98.9 | 64.9 | 90.4 | 95.9 | **528.2** |
| ***BUTD+BERT*** | | | | | | | | | | | | | | |
| DSRAN(2020)[34] | 77.8 | 95.1 | 97.6 | 59.2 | 86.0 | 91.9 | 507.6 | 78.3 | 95.7 | 98.4 | 64.5 | 90.8 | 95.8 | 523.5 |
| VSRN++*(2022)[14] | 79.2 | 94.6 | 97.5 | 60.6 | 85.6 | 91.4 | 508.9 | 77.9 | 96.0 | 98.5 | 64.1 | 91.0 | 96.1 | 523.6 |
| MV(2022)[17] | 82.1 | 95.8 | 97.9 | 63.1 | 86.7 | 92.3 | 517.5 | 80.4 | 96.6 | 99.0 | 64.9 | 91.2 | 96.0 | 528.1 |
| CHAN(2023)[22] | 80.6 | 96.1 | 97.8 | 63.9 | 87.5 | 92.6 | 518.5 | 81.4 | 96.9 | 98.9 | 66.5 | 92.1 | **96.7** | 532.6 |
| HREM*(2023)[5] | **84.0** | 96.1 | **98.6** | 64.4 | **88.0** | 93.1 | 524.2 | 82.9 | 96.9 | 99.0 | 67.1 | 92.0 | 96.6 | 534.6 |
| **DIAS(ours)** | 83.8 | 96.6 | 98.3 | **64.5** | **88.0** | **93.3** | **524.5** | 83.4 | 97.1 | 99.1 | 67.6 | 92.4 | 96.6 | **536.2** |

We combine all soft-thresholds as $\mathbf{K}^V = \{\kappa_i^V | i \in [1, N]\}$, and obtain $\mathbf{K}^T = \{\kappa_j^T | j \in [1, N]\}$ in a similar process. It is important to note that $\mathbf{K}^V$ and $\mathbf{K}^T$ are distinct. For example, image $\mathbf{V}_i$ may exhibit a high dependency degree on text $\mathbf{T}_j$, but $\mathbf{T}_j$ may not necessarily have a high dependency degree on $\mathbf{V}_i$. To avoid introducing weakly correlated information, we select spatial relationships that meet the requirements of both $\mathbf{K}^V$ and $\mathbf{K}^T$:

$$\hat{\mathbf{L}}_x = \mathbf{B}_x \mathbf{L}_x \tag{13}$$

Here $\mathbf{B}_x = \{b_{i,j}^x | i, j \in [1, N]\}$ is a binary mask matrix to save strong correlation relationships, and:

$$b_{i,j}^x = \begin{cases} 1, & |x_{i,j} - x_{j,i}| > max(\kappa_i^V, \kappa_j^T) \\ 0, & otherwise \end{cases} \tag{14}$$

$max(\cdot)$ is the function for calculating the maximum value. Base on the sparse inter-modality spatical matrix $\hat{\mathbf{L}}_x$, we update Eq.8 as:

$$\mathcal{L}_{inter} = ||\hat{\mathbf{L}}_x||_2^2 = \mathbf{B}_x ||\mathbf{X} - \mathbf{X}^\top||_2^2 = \sum_{i=1}^N \sum_{j=1}^N b_{i,j}^x (x_{i,j} - x_{j,i})^2$$
$$= \sum_{i=1}^N \sum_{j=1}^N b_{i,j}^x (\sigma_x(\hat{\mathbf{V}}_i, \hat{\mathbf{T}}_j) - \sigma_x(\hat{\mathbf{V}}_j, \hat{\mathbf{T}}_i))^2 \tag{15}$$

By performing similar operations on $\mathbf{L}_{yz}$, we can obtain the sparse intra-modality spatical matrix $\hat{\mathbf{L}}_{yz}$ and update Eq.10 as:

$$\mathcal{L}_{intra} = ||\hat{\mathbf{L}}_{yz}||_2^2 = \mathbf{B}_{yz} ||\mathbf{Y} - \mathbf{Z}||_2^2 = \sum_{i=1}^N \sum_{j=1}^N b_{i,j}^{yz} (y_{i,j} - z_{i,j})^2$$
$$= \sum_{i=1}^N \sum_{j=1}^N b_{i,j}^{yz} (\sigma_y(\hat{\mathbf{V}}_i, \hat{\mathbf{V}}_j) - \sigma_z(\hat{\mathbf{T}}_i, \hat{\mathbf{T}}_j))^2 \tag{16}$$

Here $\mathbf{B}_{yz} = \{b_{i,j}^{yz} | i, j \in [1, N]\}$.

## 3.5 Objective Function

We combine the proposed regularization terms with the triplet loss to obtain the loss function of DIAS:

$$\mathcal{L} = \mathcal{L}_{loc} + \omega_{dim}\mathcal{L}_{dim} + \omega_{inter}\mathcal{L}_{inter} + \omega_{intra}\mathcal{L}_{intra} \tag{17}$$

Here $\omega_{dim}$, $\omega_{inter}$ and $\omega_{intra}$ are hyper-parameters to control the effectiveness degree of each term. To ensure effective cross-modal interactions, we use neighbor sampling instead of random sampling for batches. First, we apply K-means [21, 29] clustering on the local image embeddings. Then, we randomly select $M$ clusters and choose $P$ images from each cluster. Finally, we pair each image with a positive text instance and obtain $N = P \times K$ image-text pairs for each batch.

## 4 EXPERIMENTS

### 4.1 Experimental Setup

**Datasets and Evaluation Metrics.** We evaluate DIAS mainly on Flickr30k [36] and MSCOCO [19] datasets. Flickr30k contains 29,000 images for training, 1,000 images for validation, and 1,000 images for testing. MSCOCO contains 123,287 images for training, 5,000 images for validation, and 5,000 images for testing. Each image of the two datasets is associated with 5 texts. The results on MSCOCO are reported on averaging over 5-folds of 1,000 test images and on the entire 5,000 test images. As a common practice in information retrieval [2], we adopt the Recall at K (R@K) to meansure the performance, and set K=1,5,10. R@K means the percentage of ground truth in the retrieved top-K lists. rSum reflects the overall matching performance, which is the sum of R@K in both image-to-text and text-to-image matching.

**Implementation Details.** We use the pre-extracted local image embeddings [2] for images, and the BiGRU [27] or BERT [4] to extract local text embeddings. All correlation algorithms default

**Table 2: Comparisons with state-of-the-art methods on MSCOCO 5K test-set. * denotes the ensemble results of two models. The bests are in bold.**

| Methods | I→T | | | T→I | | | rSum |
|---|---|---|---|---|---|---|---|
| | R@1 | R@5 | R@10 | R@1 | R@5 | R@10 | |
| ***BUTD+BiGRU*** | | | | | | | |
| GPO | 56.6 | 83.6 | 91.4 | 39.3 | 69.9 | 81.1 | 421.9 |
| MV | 56.7 | 84.1 | 91.4 | 40.3 | 70.6 | 81.6 | 424.6 |
| NAAF* | 58.9 | 85.2 | 92.0 | 42.5 | 70.9 | 81.4 | 430.9 |
| CHAN | **60.2** | 85.9 | 92.4 | 41.7 | 71.5 | 81.7 | 433.4 |
| NUIF-d | 59.3 | 85.5 | 92.0 | 41.9 | 71.3 | 81.8 | 431.8 |
| **DIAS(ours)** | 59.8 | **86.0** | **92.5** | **42.7** | **71.8** | **82.5** | **435.3** |
| ***BUTD+BERT*** | | | | | | | |
| VSRN++* | 54.7 | 82.9 | 90.9 | 42.0 | 72.2 | 82.7 | 425.4 |
| MV | 59.1 | 86.3 | 92.5 | 42.5 | 72.8 | 83.1 | 436.3 |
| CHAN | 59.8 | 87.2 | 93.3 | 44.9 | 74.5 | 84.2 | 443.9 |
| HREM* | 64.0 | 88.5 | 93.7 | 45.4 | 75.1 | 84.3 | 450.9 |
| **DIAS(ours)** | **64.4** | **88.9** | **94.1** | **47.2** | **76.5** | **85.2** | **456.3** |

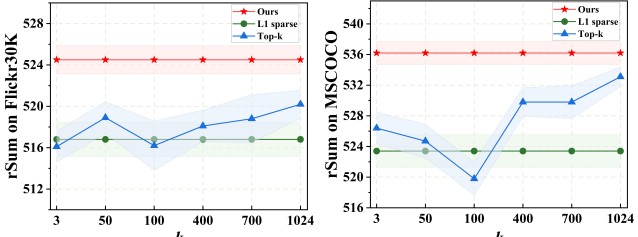

**Figure 6: The effectiveness of sparse correlation algorithm.**

to cosine similarity [28]. The experiments are conducted on an NVIDIA GeForce RTX 4090 GPU. We set 30 training epochs, and the batch size is 128 for Flickr30k and 256 for MSCOCO. Adam optimizer is adopted with an initial learning rate of $5e^{-4}$ and decaying by 10% every epochs.

## 4.2 Comparisons with State-of-the-art Methods

To verify the performance superiority of our proposed DIAS, we compare it with the state-of-the-art models on two datasets. Existing methods are divided into two types based on their feature backbones for fair comparisons. The experimental results are cited directly from respective papers. Our model reports the single model performance without the ensemble improving trick.

Quantitative results on Flickr30K and MSCOCO 1K test-sets are shown in Table 1. DIAS outperforms state-of-the-art methods with impressive margins for the R@K and rSum, and achieves consistent superiority on different textual encoders. Furthermore, Table 2 shows the more extensive database of MSCOCO 5K test-set, DIAS also performs best on nearly all metrics.

## 4.3 Ablation Study and Discussion

To demonstrate the effectiveness of components in DIAS, we conduct ablation studies on both datasets. The baseline w/o DIA means

**Table 3: Ablation studies of our model on Flickr30K and MSCOCO 1K.**

| Methods | Flickr30K | | | | MSCOCO 1K | | | |
|---|---|---|---|---|---|---|---|---|
| | I→T | | T→I | | I→T | | T→I | |
| | R@1 | R@5 | R@1 | R@5 | R@1 | R@5 | R@1 | R@5 |
| ***BUTD+BiGRU*** | | | | | | | | |
| w/o DIA | 79.3 | 94.9 | 58.9 | 84.0 | 78.9 | 95.6 | 63.0 | 90.2 |
| w/o $L_x$ | 81.1 | 95.7 | 59.5 | 84.6 | 80.4 | 96.2 | 64.2 | 90.3 |
| w/o $L_{yz}$ | 80.8 | 95.2 | 59.5 | 84.6 | 80.1 | 96.2 | 63.7 | 90.2 |
| w/o $\hat{L}_x$ | 81.6 | 96.0 | 60.2 | 84.8 | 81.2 | 96.5 | 64.7 | 90.3 |
| w/o $\hat{L}_{yz}$ | 81.5 | 95.8 | 59.9 | 84.7 | 80.9 | 96.4 | 64.1 | 90.2 |
| **DIAS** | 81.8 | 96.1 | 60.7 | 84.9 | 81.3 | 96.8 | 64.9 | 90.4 |
| ***BUTD+BERT*** | | | | | | | | |
| w/o DIA | 80.8 | 95.5 | 62.9 | 85.9 | 80.7 | 96.1 | 65.1 | 91.1 |
| w/o $L_x$ | 83.3 | 96.2 | 64.4 | 87.8 | 82.9 | 97.0 | 66.9 | 92.1 |
| w/o $L_{yz}$ | 82.7 | 95.9 | 63.7 | 87.2 | 82.1 | 96.8 | 66.3 | 91.8 |
| w/o $\hat{L}_x$ | 83.5 | 96.2 | 64.4 | 87.9 | 83.0 | 97.1 | 67.2 | 92.2 |
| w/o $\hat{L}_{yz}$ | 83.4 | 96.2 | 64.0 | 87.8 | 82.8 | 97.0 | 67.0 | 92.2 |
| **DIAS** | 83.8 | 96.6 | 64.5 | 88.0 | 83.4 | 97.1 | 67.6 | 92.4 |

**Table 4: The effect of applying dimension information alignment (abbreviated as *DIA*) to other models.**

| Methods | Flickr30K | | | | MSCOCO 1K | | | |
|---|---|---|---|---|---|---|---|---|
| | I→T | | T→I | | I→T | | T→I | |
| | R@1 | R@5 | R@1 | R@5 | R@1 | R@5 | R@1 | R@5 |
| MV | 82.1 | 95.8 | 63.1 | 86.7 | 80.4 | 96.6 | 64.9 | 91.2 |
| **+DIA** | 82.9 | 96.2 | 63.8 | 87.2 | 81.4 | 96.8 | 65.8 | 91.9 |
| CHAN | 80.6 | 96.1 | 63.9 | 87.5 | 81.4 | 96.9 | 66.5 | 92.1 |
| **+DIA** | 82.0 | 96.4 | 64.2 | 87.8 | 81.7 | 96.9 | 66.9 | 92.3 |
| HREM | 84.0 | 96.1 | 64.4 | 88.0 | 82.9 | 96.9 | 67.1 | 92.0 |
| **+DIA** | 84.2 | 96.5 | 64.6 | 88.0 | 83.0 | 97.2 | 67.5 | 92.5 |

DIAS without dimention information alignment. w/o $L_x$ and w/o $L_{yz}$ denote the lack of inter- and intra-modality spatial constraints, respectively. w/o $\hat{L}_x$ and w/o $\hat{L}_{yz}$ mean that no sparsity regularization is applied on inter- and intra-modality spatial matrices, respectively. According to the results shown in Table 3, we have the following observations:

(1) **The effectiveness of model designing.** Removing any components in DIAS reduced performance, which indicates the proposed dimension information alignment, spatial constraints, and sparse correlation algorithm are effective for image-text matching tasks.

(2) **Discussion on dimension information alignment.** The performance of w/o DIA is the worst among the baselines, indicating that aligning dimension information is the most crucial component for DIAS. To further discuss the effectiveness of this component, we apply it to other models. The results shown in Table 4 demonstrate dimension information alignment can also improve the performance of other models to a certain extent.

(3) **Discussion on spatial constraint.** The performance of w/o $L_{yz}$ is inferior to w/o $L_x$, indicating that introducing intra-modality

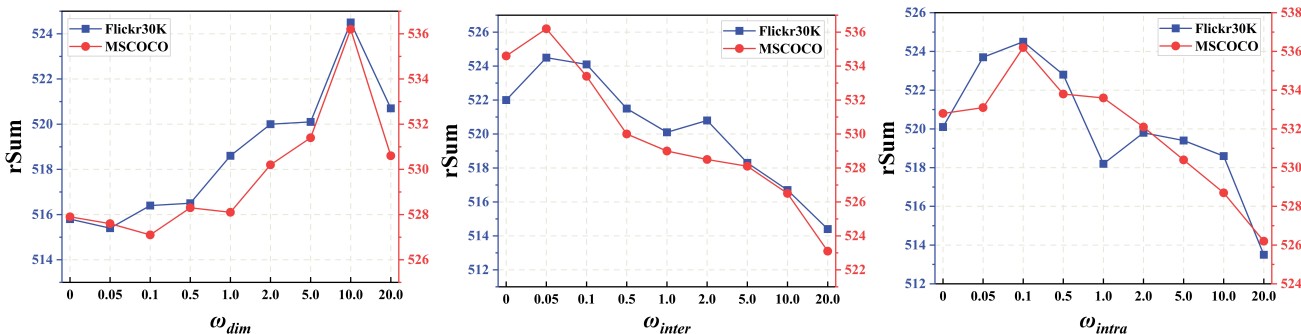

**Figure 7: Performance comparison on varying.**

spatial constraint is more effective for DIAS than inter-modality constraint. This result provides evidence for the viewpoint that intra-modality constraint is not affected by modality gap and can directly assist the model in learning robust features.

(4) **Discussion on sparse correlation algorithm.** The performance of w/o $\hat{L}_x$ and w/o $\hat{L}_{yz}$ are inferior to the complete DIAS, suggesting sparse correlation algorithm can assist the model in learning significant features by selecting strong correlated relationships. To further discuss the effectiveness of this algorithm, we compared it with the Top-k strategy and L1 sparse strategy. The Top-k strategy retains the top-k most relevant relationships for each instance. The L1 sparse strategy constrains the correlation matrix using the L1-norm. The results as shown in Fig. 6 reveal the sparse correlation algorithm outperforms these two baselines.

### 4.4 Robustness Analysis

**Parameter sensitivity.** We aim to understanding how our model performs by varying the values of hyper-parameters $\omega_{dim}$, $\omega_{inter}$ and $\omega_{intra}$, as shown in Fig.7. When varying any of these hyper-parameters, we fix others with default settings. $\omega_{dim}$, $\omega_{inter}$ and $\omega_{intra}$ obtain optimal results at 10, 0.05, and 0.1, respectively.

**Generalization study.** To validate the generalization capability of DIAS in learning latent semantics, we conduct cross-validation experiments following [40]. Specifically, we use the model trained on MSCOCO dataset to evaluate its zero-shot transferability on Flickr30K test-set. The result shown in Table 5 indicates our proposed DIAS exhibits stronger generalization ability than the baseline, confirming DIAS is capable of learning cross-modality latent semantics.

### 5 CONCLUSION

This paper proposes a novel image-text matching model based on dimension information alignment and sparse spatial correlation algorithm (DIAS). We explicitly align information representation of embeddings in corresponding dimension, to address the issue of lack of rationality in correlation calculation caused by modality gap. Additionally, by introducing inter- and intra-modalities spatial relationships, we enhance the constraints during the cross-modal interaction. More importantly, we propose a sparse correlation algorithm to select strong spatial relationships to reduce the requirement for symmetric of embeddings, allowing the model to

focus on learning more significant structural features. Extensive experiments and analyses conducted on two datasets show the superiority and rationality of DIAS.

**Table 5: Generalization ability comparison of models trained on MSCOCO and validated on Flickr30K test-set.**

| | I→T | | | T→I | | | rSum |
|---|---|---|---|---|---|---|---|
| | R@1 | R@5 | R@10 | R@1 | R@5 | R@10 | |
| ***BUTD+BiGRU*** | | | | | | | |
| Baseline | 53.2 | 82.1 | 88.7 | 42.5 | 71.1 | 79.5 | 417.1 |
| **DIAS** | 69.2 | 91.2 | 95.0 | 54.5 | 79.4 | 87.0 | 476.3 |
| ***BUTD+BERT*** | | | | | | | |
| Baseline | 60.6 | 85.4 | 91.4 | 46.7 | 73.7 | 81.8 | 439.6 |
| **DIAS** | 73.9 | 92.2 | 96.2 | 57.6 | 80.8 | 87.6 | 488.3 |

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
