# OpenReview forum: "Bridging the Modality Gap: Dimension Information Alignment and Sparse Spatial Constraint for Image-Text Matching"
_acmmm.org/ACMMM/2024/Conference — MM2024 Poster_

### Official Review · Reviewer_53Zr · 2024-05-20

**Rating:** 4
**Confidence:** 4

**Summary:**

This paper proposes the dimension information alignment and sparse spatial correlation framework for image-text matching task. This framework aligns information representation of embeddings in corresponding dimension to address the issue of lack of rationality in correlation calculation in cross-modality semantics. Then It proposes a sparse correlation module to select strong spatial relationships to reduce the requirement for symmetric of embeddings. Extensive experiments conducted on two datasets show the superiority of the proposed framework.

**Strengths:**

1. This paper proposes a dimension information alignment method to enhance the rationality of cross-modal interaction and suppress feature redundancy.
2. This paper proposes a correlation algorithm to select strong correlated spatial relationships and reduce the need for symmetry of embeddings.
3. The cross-modal retrieval experiments on Flickr and MSCOCO show the effectiveness of the proposed framework.
4. This paper is easy to read.

**Limitations:**

1. The first concern is the motivation of this paper. The core idea of the paper is to make the constraint between intra-modal semantic consistency (image-image, text-text) and inter-modal semantic semantic consistency (image-text), as shown in Fig 1. The idea is too common and has been explored in many image-text matching [1-5] papers. Besides, the images and texts contain the semantic gap in nature, directly applying the intra-modal semantic relationship (e.g., the similarity/distance matrix) to supervise the intra/inter-modal semantic relationship is an unclear method.  This paper requires more theoretical and experimental analysis for the semantic consistency, e.g., a rigorous definition of consistency.
2. The experimental results are not convincing. This paper only contains the results on Faster-RCNN, which are out-of-date in 2020s. This paper should choose the Vision Transformer as visual encoders to extract patch features, rather than the abandoned region features, hence I suggest the authors using the CLIP models as the backbone, it will be a great choice.
3. Existing models in various multi-modal learning fields (also including image-text matching) tend to be unified transformer modeling + big data pre-training.  I understand that not all researchers can do experiments on visual-language pre-training, but I hope to see small experiments that can enlighten the pre-training and push the image-text matching field forward, rather than modifying the network and hyper-parametrizing training, especially now that large language models (e.g., ChatGPT) and large multimodal models (e.g., GPT-4V) has entered public life. What do the authors think this paper can do to inspire existing pre-training? It is suggested that these unique heuristic points can be added in the paper.


[1] Ladder Loss for Coherent Visual-Semantic Embedding, AAAI2020

[2] Adaptive Offline Quintuplet Loss for Image-Text Matching, ECCV2020

[3] Universal Weighting Metric Learning for Cross-Modal Matching, CVPR2020

[4] Deep Evidential Learning with Noisy Correspondence for Cross-modal Retrieval, ACM MM, 2022

[5] An Image Worth Five Sentences A New Look into Semantics for Image-Text Matching, WACV, 2022

**Suitability:**

3

---

### Official Review · Reviewer_TGP9 · 2024-05-24

**Rating:** 4
**Confidence:** 3

**Summary:**

This paper belongs to the field of cross-modal image-text retrieval, aiming at solving the feature differences between different modalities, this paper proposes a DIAS algorithm, which mainly solves related problems from the alignment of the corresponding dimension information of features and the spatial constraints inside and outside the modality.

**Strengths:**

1. A dimension information alignment method for different modal embeddings is proposed to improve the rationality of cross-modal interaction and suppress feature redundancy.
2. New inter-modal and intra-modal constraints are introduced to ensure the effectiveness of semantic alignment.
3. A sparse correlation algorithm is proposed to select strongly correlated spatial relations, which reduces the requirement for embedding symmetry.

**Limitations:**

1. The meaning expressed in Figure 1 is not very clear, and the intention is not clearly expressed.
2. Is $v^{i}$ calculated in Formula 1 the attend vector in the SCAN paper? HREM in Table 4 does not have much improvement effect after dimension information alignment is added. What is the reason?
3. The paper format is not in accordance with the requirements of ACM MM 2024

**Suitability:**

3

---

### Official Review · Reviewer_hCDT · 2024-05-26

**Rating:** 4
**Confidence:** 4

**Summary:**

The article introduces a method called DIAS (Dimensional Information Alignment and Sparse Space Constraint) designed to address the cross-modal gap in image-text matching. This paper proposes a novel method using dimensional information alignment and sparse space constraints to mitigate the modality gap of text and image. moreover, a sparse correlation algorithm is introduced to select strongly correlated spatial relationships, enabling the model to learn more significant features and avoid being misled by weak correlation. Extensive experiments showed the effectiveness of the proposed method.

**Strengths:**

(1)The author introduces a novel method using dimensional information alignment and sparse space constraints to mitigate the modality gap of text and image.

(2)The method description is clear, and the logic is reasonable.

(3)Experimental results on standard benchmarks demonstrating improvements in performance metrics of the proposed method.

**Limitations:**

(1)How did previous work address the issue of dimensional misalignment between modalities? What is the fundamental difference between these approaches and the method proposed in this paper? The motivation of this paper seems not very clear.

(2)Why is it necessary to learn a soft-threshold? Does it imply that parts below this threshold are not important?

(3)The authors claim that the proposed method is efficient with respect to computational consumption, but the paper lacks a clear analysis of this aspect. It is suggested that this paper provide further experimental verification and theoretical analysis on this.

**Suitability:**

3

---

### Meta-Review · Area_Chair_uY3p · 2024-07-01

**Recommendation:** Accept (Poster)
**Confidence:** 5

**Metareview:**

This paper proposes to use dimensional information alignment and sparse space constraints to mitigate the modality gap between text and images. The motivation is clear, the designed modules have originality, and the experimental results are promising.
All reviewers agree that the paper has its own innovation and have also raised concerns about the experiments.
After carefully examining all the reviewers' comments, the manuscript and the rebuttal, the ACs believe that there are some issues with the experiments.
In the experiment using BUTD + BiGRU, the authors compared with NUIF-d,
but did not do so when using BUTD+BERT, whereas NUIF-d performs significantly better (Flickr30K: 524.5 vs 529.4, MSCOCO-1K: 536.2 vs 538.2, MSCOCO-5k: 456.3 vs 459.1).
Certainly, a method does not always have to achieve the best performance, but the comparisons should be open and sincere.
Considering that the proposed Dimension Information Alignment is innovative and its effectiveness has also been demonstrated through combination with other methods, the ACs believe that this paper can provide some inspiration to the field and have decided to accept it.
The authors need to carefully refine the experiments and analysis based on the feedback from the reviewers and ACs.